# Organic-to-Aqueous Phase Transfer of Alloyed AgInS_2_-ZnS Nanocrystals Using Simple Hydrophilic Ligands: Comparison of 11-Mercaptoundecanoic Acid, Dihydrolipoic Acid and Cysteine

**DOI:** 10.3390/nano11040843

**Published:** 2021-03-25

**Authors:** Patrycja Kowalik, Piotr Bujak, Mateusz Penkala, Adam Pron

**Affiliations:** 1Faculty of Chemistry, Warsaw University of Technology, Noakowskiego 3, 00-664 Warsaw, Poland; pkowalik@ch.pw.edu.pl (P.K.); apron@ch.pw.edu.pl (A.P.); 2Faculty of Chemistry, University of Warsaw, Pasteura 1 Str., PL-02-093 Warsaw, Poland; 3Institute of Chemistry, Faculty of Mathematics, Physics and Chemistry, University of Silesia, Szkolna 9, 40-007 Katowice, Poland; mateusz.penkala@us.edu.pl

**Keywords:** AgInS_2_-ZnS alloyed nanocrystals, hydrophilic ligands, ligand exchange, 11-mercaptoundecanoic acid, dihydrolipoic acid, cysteine

## Abstract

The exchange of primary hydrophobic ligands for hydrophilic ones was studied for two types of alloyed AgInS_2_-ZnS nanocrystals differing in composition and by consequence exhibiting two different emission colors: red (R) and green (G). Three simple hydrophilic ligands were tested, namely, 11-mercaptoundecanoic acid, dihydrolipoic acid and cysteine. In all cases, stable aqueous colloidal dispersions were obtained. Detailed characterization of the nanocrystal surface before and after the ligand exchange by NMR spectroscopy unequivocally showed that the exchange process was the most efficient in the case of dihydrolipoic acid, leading to the complete removal of the primary ligands with a relatively small photoluminescence quantum yield drop from 68% to 40% for nanocrystals of the R type and from 48% to 28% for the G ones.

## 1. Introduction

Colloidal nanocrystals of inorganic semiconductors attract the attention of materials science and materials chemistry communities mainly because of their tunable photo- and electroluminescence properties [1]. Among a plethora of binary, ternary and quaternary nanocrystals of different types, alloyed AgInS_2_-ZnS ones deserve special interest for two main reasons: (i) they do not contain toxic elements such as previously studied cadmium or lead chalcogenides, which makes them suitable for diversified biomedical applications; (ii) their photoluminescence spectrum can be precisely tuned by adjusting their composition. Moreover, in recent years, several efficient preparation procedures were elaborated by various research groups, leading to alloyed AgInS_2_-ZnS nanocrystals exhibiting high photoluminescence quantum yield (PLQY) values [2,3,4,5,6,7,8,9].

For many applications, including biomedical ones, nanocrystals of hydrophilic nature are needed, capable of forming stable aqueous colloidal solutions. Two strategies can be envisioned in the preparation of such nanocrystals [4,10]. In the first one, the nanocrystals are formed in aqueous solutions and their colloidal stability in this medium is assured by capping ligands of hydrophilic nature. The second one consists of preparing nanocrystals in non-aqueous media where they are stabilized with hydrophobic ligand. In the second step, the exchange of hydrophobic ligands for hydrophilic ones is performed, leading to the nanocrystals’ transfer to water, where they form stable dispersions. However, taking into account the possible versatility of the surface modification, the second method seems more appropriate. This especially applies to hybrid systems frequently containing complicated bio-active molecules. The use of hydrophilic primary ligands severely limits the number of possible surface modification pathways as well as ligand exchange procedures [4]. Moreover, it sometimes makes the introduction of biologically active molecules difficult.

As already mentioned, initial research on highly luminescent hydrophilic semiconductor nanocrystals was focused on binary cadmium or lead chalcogenides whose emission spectrum was tuned either by alloying with other binary semiconductors or by the quantum confinement effect [11]. However, in the majority of biomedical applications, the toxicity of cadmium constituted a real problem, especially for in vivo experiments. The leaching of cadmium cations to the environment could frequently be observed, despite depositions of protective layers of different types [12,13,14]. For these reasons, intensive research started, aimed at the preparation of free of toxic cadmium, ternary (AgInS_2_, AgInS_2_) and quaternary (AgInS_2_-ZnS and CuInS_2_-ZnS) nanocrystals either alloyed or core/shell type. Recent investigations demonstrated that their PLQY values could match those measured for cadmium chalcogenides [2,3,4,5,6,7,8,9]. Hydrophilic AgInS_2_-ZnS nanocrystals capped with bioactive ligands are especially interesting because in recent years they have been extensively tested in anti-cancer therapy [15,16,17,18,19,20,21].

We shared this interest working in the past decade on stoichiometric (AgInS_2_) and nonstoichiometric (Ag-In-S) ternary nanocrystals as well as alloyed quaternary ones (AgInS_2_-ZnS) [22,23,24,25]. We also tested them in anti-cancer therapies [17,18,20,21]. From this perspective, the elaboration of efficient ligand exchange procedures, facilitating the transfer of the modified nanocrystals to water, is of crucial importance. In this paper, we comparatively describe the ligand exchange-induced hydrophilization of two types of alloyed AgInS_2_-ZnS nanocrystals differing in composition and by consequence emitting light of different colors: red (**R**) and green (**G**). Three popular hydrophilic ligands were tested: 11-mercaptoundecanoic acid (MUA), dihydrolipoic acid (DHLA) and L-cysteine (Cys). Detailed characterization of nanocrystals and their surface was performed prior to the ligand exchange and after the exchange, which involved the determination of their size and shape, composition, crystal structure and luminescent properties. NMR was used for the identification of primary ligands and the ligands introduced upon the exchange.

## 2. Materials and Methods

### 2.1. Materials

Silver nitrate (99%), indium(III) chloride (98%), zinc stearate (technical grade), 1-dodecanethiol (DDT, 98%), sulfur (99%), 1-octadecene (ODE, 90%), oleylamine (OLA, 70%), 11-mercaptoundecanoic acid (MUA, 95%), lipoic acid (LA, 99%), NaBH_4_ (96%), L-cysteine (Cys, 96%), CDCl_3_ (100%, 99.96 atom% D), D_2_O (100%, 99.99 atom% D), and NaOD (40 wt. % in D_2_O, 99.5 atom% D) were supplied by Sigma-Aldrich (Merk KGaA, Darmstadt, Germany).

### 2.2. Preparation of AgInS_2_-ZnS Nanocrystals (R-“Red” and G-“Green” Types)

Initial hydrophobic AgInS_2_-ZnS nanocrystals (**R** and **G**) were synthesized by a modification of the method described in literature [22,23]. Detailed description of the applied procedure can be found in Appendix A.

### 2.3. Primary Ligand Exchange for 11-Mercaptoundecanoic Acid (MUA) 

All operations were carried out under a constant dry argon flow. A mixture of MUA (0.5 g, 2.3 mmol) and NaOH (0.1 g, 2.5 mmol) in water (10 mL) was heated with stirring at 50 °C until a homogenous solution was formed. Then, a toluene dispersion (10 mL) of nanocrystals (**R** or **G**) prepared as described in Appendix A was injected into this solution. The as-obtained two-phase mixture was heated at 80 °C for 8 h under argon. After cooling, the reaction mixture was centrifuged to obtain a complete phase separation; the solid and the organic phase were discarded. The aqueous phase was then mixed with 20 mL of acetone which led to the precipitation of **R-MUA** (or **G-MUA**) nanocrystals. After centrifugation, the nanocrystals were redispersed in 10 mL of water.

### 2.4. Primary Ligand Exchange for Dihydrolipoic Acid (DHLA)

All operations were carried out under a constant dry argon flow. To a solution of lipoic acid (0.2 g, 0.97 mmol) and NaOH (0.05 g, 1.25 mmol) in water (10 mL), sodium borohydride (0.08 g, 2.11 mmol) was added and the resulting mixture was stirred at 40 °C under argon for 1 h. Then, a toluene dispersion (10 mL) of nanocrystals (**R** or **G**) prepared as described in Appendix A was injected into this solution. The mixture was stirred at 40 °C for 4 h and additionally at room temperature for 8 h. Then, it was centrifuged to obtain a complete phase separation and the organic phase was discarded. The remaining aqueous phase was then mixed with 20 mL of acetone which led to the precipitation of **R-DHLA** (or **G-DHLA**) nanocrystals. After centrifugation, the nanocrystals were redispersed in 10 mL of water.

### 2.5. Primary Ligand Exchange for L-Cysteine (Cys)

All operations were carried out under a constant dry argon flow. A mixture of L-cysteine (3.0 g, 24.8 mmol), NaOH (0.05 g, 1.25 mmol) in water (10 mL) was heated with stirring at 40 °C until a homogenous solution was formed. Then, the toluene dispersion (10 mL) of nanocrystals (**R-Cys** or **G**) prepared as described in Appendix A was injected into this solution. The as-obtained two-phase mixture was heated at 40 °C for an additional 4 h under argon. After cooling, the reaction mixture was centrifuged to obtain a complete phase separation, and the organic phase was discarded. The remaining aqueous phase was then mixed with 20 mL of acetone which led to the precipitation of **R-Cys** (or **G-Cys**) nanocrystals. After centrifugation, the nanocrystals were redispersed in 10 mL of water. 

### 2.6. Characterization

Elemental analysis was performed with a multichannel Quantax 400 energy-dispersive X-ray spectroscopy (EDS) system with a 125 eV xFlash detector 5010 (Bruker, Billerica, MA, USA) using a 15 kV electron beam energy. Transmission electron microscopy (TEM) studies were carried out on a Zeiss Libra 120 electron microscope operating (Carl Zeiss, Oberkochen, Germany) at 120 kV. Emission spectra were recorded using a Hitachi F-7000 spectrophotometer (Hitachi, Tokyo, Japan) with a standard 10 mm cell. The quantum yields of luminescence were determined by the absolute method at room temperature, using an integrating sphere with solvent as a blank. 1H, and 1H-1H COSY NMR spectra were recorded on a Bruker Avance (400 MHz) spectrometer (Bruker, Billerica, MA, USA) and referenced with respect to tetramethylsilane (TMS) and solvents.

## 3. Results and Discussion

The ligand exchange described in this research was carried out for two types of alloyed AgInS_2_-ZnS nanocrystals, termed **R**
*(red)* and **G**
*(green),* respectively, differing in composition and by consequence emitting light of different wavelengths. These nanocrystals were prepared through the injection of sulfur dissolved in oleylamine (S/OLA) to a mixture of AgNO_3_, InCl_3_, zinc stearate, and 1-dodecanothiol (DDT) dissolved in 1-octadecene (ODE). This procedure allows for obtaining alloyed nanocrystals of orthorhombic AgInS_2_—hexagonal ZnS smoothly varying in composition via strict control of the precursor molar ratios [23,24]. As a result, their luminescence can be precisely tuned over essentially the entire range of the visible spectrum, still retaining high photoluminescence quantum yield (PLQY) values.

**R** and **G** nanocrystals were spherical in shape. Their diameters, as determined from TEM images, were 6.2 ± 0.9 nm and 4.2 ± 0.6 nm, respectively (Figure 1). The results of EDS analysis led to the following empirical formulae: Ag_1.00_In_2.80_Zn_1.30_S_4.00_(S_6.00_) (**R**) and Ag_1.00_In_1.50_Zn_7.80_S_17.0_(S_10.50_) (**G**) (see Appendix A). The emission spectra of both types of nanocrystals showed a rather broad peak at 720 nm (**R**) and 543 nm (**G**) (see Figure 2). High values of their PLQYs should be noted (67% (**R**) and 58% (**G**)) and for this reason they were selected for the ligand exchange investigations. All microscopic and spectroscopic data of the studied nanoparticles are collected in Table 1.

For the identification of primary ligands, we used a procedure previously elaborated in our group and applied with success to nanocrystals of various types [26]. It consists of dissolving the inorganic core followed by extraction of the organic residue which is then investigated by ^1^H NMR. Application of this procedure to **R** and **G** nanocrystals allowed for unequivocal identification of two types of ligands: stearic acid anions bound to zinc cations and 1-aminooctadecane co-ordinated to indium cations. The latter is a product of the hydrogenation of OLA occurring in the course of the nanocrystals’ synthesis [24].

Three simple and popular hydrophilic ligands were selected for the investigation of the ligand exchange process, namely, 11-mercaptoundecanoic acid (MUA), dihydrolipoic acid (DHLA) and L-cysteine (Cys) (Scheme 1). 

Ligands were exchanged in one step. In a typical procedure, dispersions of **R** or **G** nanocrystals in toluene were added to a saturated alkaline solution of either **MUA** or **DHLA** or **Cys**. The resulting mixtures were vigorously stirred for 4 to 8 h. Then, the nanocrystals were precipitated with acetone and separated by centrifugation. Finally, they were redispersed in water. The chemical compositions of the resulting nanocrystals (**R-MUA**, **R-DHLA**, **R-Cys**, **G-MUA**, **G-DHLA**, **G-Cys**) were determined by EDS (for the corresponding spectra see Appendix A). Their extended characterization data are collected in Table 1.

11-Mercaptoundecanoic acid (MUA) seems to be the most popular ligand used for rendering binary and ternary nanocrystals hydrophilic via the exchange of hydrophobic primary ligands [27,28]. At room temperature, it is practically insoluble in water. Thus, the exchange of ligands had to be carried out at 50 °C in a saturated solution of MUA formed by mixing 500 mg of MUA and 100 mg of NaOH in 10 mL of water.

In Figure 3, the ^1^H NMR spectrum of MUA dissolved in NaOD/D_2_O (pH ~10) is compared with the spectra of aqueous dispersions of **R-MUA** and **G-MUA**. Careful analysis of the spectra registered for nanocrystal dispersions in the spectral range of signals originating from the methylene groups of long aliphatic chains (1.2–1.5 ppm) seems to indicate incomplete exchange of primary ligands for MUA. This effect is more pronounced in the spectrum of **R-MUA**. 

Before discussing in detail the spectra of **R-MUA** and **G-MUA** dispersions, it is instructive to describe specific features of the spectrum of “free” **MUA**. Two distinctive triplets can be distinguished in it: at 2.13 and 2.49 ppm. They correspond to methylene protons in α position with respect to -COOH and -SH terminal groups. In the range of 1.47–1.59 ppm, overlapping multiplets originating from β-methylene protons are observed. In the spectrum of the dispersion of **R-MUA**, a slight shift of the triplet corresponding to -CH_2_COOH is observed from 2.13 to ~2.16 ppm. Significantly more pronounced changes are observed in the case of the signals originating from to -CH_2_SH. The presence of three poorly resolved co-existing triplets can be distinguished at 2.49, 2.68 and 2.77 ppm which can be interpreted in terms of three non-equivalent co-ordination sites imposing different conformations of the bound ligands. Large shifts of the signal corresponding to -CH_2_CH_2_SH additionally corroborate the binding of MUA to the nanocrystal surface. The same effect was previously observed in the case of primary ligands binding to the surface of growing Ag_2_S and AgInS_2_ nanocrystals [25]. The presence of only one triplet attributable to -CH_2_SH in the spectrum of the dispersion of **G-MUA** indicates that all **MUA** are in this case spectroscopically equivalent. All these attributions were verified through the analysis of ^1^H-^1^H COSY spectra presented in Appendix A (Appendix A).

Chemical composition of nanocrystals before and after the exchange of ligands can be determined from the EDS data. First, it provides information on the molar ratio of elements constituting the inorganic core. Second, it allows for the estimation of the organic part contribution to the total mass of ligand-capped nanocrystals. Carbon is the diagnostic element here because it is present only in the organic part. A small drop in the content of carbon is observed after the ligand exchange, i.e., the transformation of **R** into **R-MUA** and **G** into **G-MUA** (see Table 1). This decrease in the content of carbon is caused by replacing longer aliphatic chains (C_18_) of primary ligands in **R** and **G** by shorter ones (C_11_) in **R-MUA** and **G-MUA**. This drop is lower than expected, indicating incomplete ligand exchange, consistent with the obtained NMR data (*vide supra*).

As judged from elemental analysis, ligand exchange induces a decrease in the content of indium. For **G-MUA**, a drop in the content of zinc is additionally observed. The case of sulfur is more complex. In **R** and **G**, all sulfur originates from the inorganic core since the primary ligands do not contain this element. Upon ligand exchange, organic sulfur is introduced, and in **R-MUA** and **G-MUA** organic and inorganic forms of this element are present. Irrespective of this, after the exchange of ligands, the content of sulfur significantly exceeds the value expected from the molar ratio of metals in the inorganic core. This excess could in principle be assigned to the ligand-originating sulfur.

Although ligand exchange induces significant changes even in the inorganic core composition, they have a rather small effect on the nanocrystal size, which is diminished by less than 10% for **R-MUA** and nearly 11% in the case of **G-MUA**. The exchange procedure has the most pronounced effect of PLQY values, which drop from 67% in ***R*** to 30% in **R-MUA** and from 48% in ***G*** to 25% in **G-MUA**, still remaining among the highest reported for aqueous dispersions of semiconductor nanocrystals.

Dihydrolipoic acid (DHLA) is a popular bidentate hydrophilic ligand used for the stabilization of colloidal nanocrystals of inorganic semiconductors [29,30]. More frequently, it constitutes a binding segment of more complex ligands, frequently of macromolecular nature [31,32,33,34,35,36,37]. DHLA and its derivatives can be relatively easily obtained from commercially available lipoic acid (LA) through its reduction with NaBH_4_ [38,39]. Alternatively, it can be obtained photochemically through UV irradiation (300 nm < λ < 400 nm) [36]. 

The procedure of primary ligand exchange in alloyed AgInS_2_-ZnS nanocrystals for DHLA, elaborated for this research, consists of two steps. In the first step, DHLA is generated in situ in a mixture of LA (200 mg) and NaOH (50 mg) in water to which NaBH_4_ (80 mg) is added. The reduction is carried out at 40 °C for 1 h. LA is sparingly soluble in water but the solubility of its reduction product (DHLA) is much better. In the second step, **R** (or **G**) nanocrystal dispersions are added and the ligand exchange is carried out in the same manner as in the case of MUA ligands. For NMR studies, all these reactions were carried out in D_2_O.

In Figure 4, the ^1^H NMR spectrum of LA in CDCl_3_ is compared with the spectrum of a sample taken from the reaction medium saturated with DHLA prior to the addition of **R** (or **G**) nanocrystals. The remaining two spectra were registered for **R-DHLA** and **G-DHLA**, i.e., nanocrystals dispersed in D_2_O after ligand exchange. The analysis of the ^1^H-^1^H COSY spectrum (Appendix A) facilitates unequivocal attribution of signals originating from LA. In particular, multiplets in the spectral ranges 3.53–3.60 ppm and 3.08–3.21 ppm should be attributed to protons of -CHS- and -CH_2_S groups, respectively, whereas a triplet at 2.37 ppm originates from protons of the methylene group adjacent to the carboxylic one. Increased solubility in D_2_O of DHLA formed in situ in the mixture of LA and NaBH_4_ made the registration of its spectrum possible. The signals observed in the spectra ranges of 3.56–3.70 ppm and 3.07–3.24 ppm are broadened. They can be ascribed to protons of the following groups -CHSH and -CH_2_SH. A clear triplet at 2.11 pm originates from protons of the methyl group adjacent to the carboxylic one, (-CH_2_COOH), whereas a multiplet at 1.33–1.38 ppm is attributed to thiol groups -SH [31], which is corroborated by couplings in this spectral range of the ^1^H-^1^H COSY spectrum (Appendix A). The presence of the reducing agent (NaBH_4_) is spectroscopically manifested by two multiplets at ca. −0.2 ppm, namely, a quartet (1:1:1:1; *J* = 80.6 Hz) at ca. −0.2 ppm and a septet (1:1:1:1:1:1:1; *J* = 27 Hz) originating from ^1^H and ^11^B (I = 3/2) and ^10^B (I = 1). The clear difference in their intensities has its origin in the different abundance of ^11^B and ^10^B isotopes (80.22% and 19.78%, respectively).

Aqueous dispersions of **R-DHLA** are stable. Moreover, contrary to the case of **R-MUA**, in their ^1^H NMR, no signals in the spectral range of 1.0–1.3 ppm, attributable to the presence of primary ligands, can be observed. Thus, in this case, the ligand exchange process is efficient and complete. The broadening of spectral lines of free DHLA generated in situ (Figure 4b) is caused by its limited solubility in the reaction medium. To the contrary, spectral lines originating from **R-DHLA** nanocrystals are narrow and well resolved, indicating very good dispersibility of this system in water, indicating that aggregation of nanoparticles does not take place. A similar effect was previously observed for nanocrystals of the same core composition to which electroactive ligands were attached which improved their dispersibility [40]. 

All signals expected for the DHLA ligand are present in the ^1^H NMR spectrum of **R-DHLA**. Binding of DHLA ligands to the nanocrystal surface is manifested by small shifts of selected signals as compared to their position in the spectrum of “free” **DLHA**. In the spectral ranges 3.74–3.81 ppm and 3.21–3.33 ppm, multiplets are observed originating from protons attached to carbons directly linked to the thiol groups, i.e., -CHSH and -CH_2_SH. The signal at 2.21 ppm corresponds to the methylene group adjacent to the carboxylic one, as revealed by the ^1^H-^1^H COSY spectrum (see Appendix A). It strongly overlaps with the line ascribed to acetone, which was used for the precipitation of **R-DHLA** nanocrystals in the process of the ligand exchange. Lines clearly visible in the spectral range 1.44–1.51 ppm originate from protons of the thiol groups. The corresponding spectrum of **G-DHLA** is very similar; however, in the spectral ranges of 1.0–1.1 ppm and 3.5–3.6 ppm, i.e., diagnostic of protons of SH and -CHSH groups, additional multiplets are present, suggesting different conformations of ligands bound to the nanocrystal surface.

The exchange of primary ligands for **DHLA** results in significant changes in the elemental analysis (see Table 1). The content of carbon drops from 57.6% in **R** to 8.7% in **R-DHLA** and from 55.3% in ***G*** to 20.9% **G-DHLA**. This finding clearly indicates that in this case the ligand exchange process is very efficient and no initial ligands are present in the sample, consistent with the results of NMR spectroscopy. The exchange process also leads to a modification of the inorganic core composition manifested in **R-DHLA** as well as in **G-DHLA** by a distinct decrease in indium content and less pronounced changes in zinc content. As in the case of nanocrystals capped with MUA, the new ligands in **R-DHLA** and **G-DHLA** introduce organic sulfur in addition to the inorganic one already present in **R** and **G**; as a result, the content of sulfur significantly increases after the ligand exchange.

Changes in the chemical composition of **R-DHLA** nanocrystals, as compared to **R**, are accompanied by distinct diminution of their size. This is spectroscopically reflected by a hypsochromic shift of their emission band from 720 to 672 nm originating from the quantum confinement effect (see Figure 2). Size reduction is much less pronounced in the case of the transformation of **G** into **G-DHLA**. As a result, no quantum confinement effect can be observed in this case and the emission bands of **G** and **G-DHLA** nanocrystals are located at essentially the same positions (~540 nm) (Figure 2).

Finally, it should be stated that transforming hydrophobic **R** into hydrophilic **R-DHLA** allows for retaining the PLQY value at the relatively high level of 40%. In the case of **G-DHLA**, this value is 28%. 

L-Cysteine (in the subsequent text abbreviated as Cys) is a hydrophilic molecule frequently used as a primary ligand in the syntheses of colloidal semiconductor nanocrystals carried out in aqueous media. It is less frequently used in processes of nanocrystal hydrophilization through ligand exchange [4]. Cysteine is more soluble in water than **MUA** and **DHLA**. Apart from obvious advantages, this good solubility brings some difficulties in the preparation of solutions used for the ligand exchange. The efficiency of this process requires the application of the highest possible gradient of the ligands to be introduced, which can be assured by the use of saturated solutions. In the case of **Cys**, this solution is very concentrated. The same solutions were used in the ligand exchange process. Dispersions of **R** (or **G**) in toluene were mixed with 3.0 g of L-cysteine and 50 mg NaOH dissolved in 10 mL of water and then vigorously stirred at 40 °C for 4 h. After precipitation in acetone, the resulting **R-Cys** and **G-Cys** could be readily redispersed in water.

In Figure 5, the ^1^H NMR spectra of the free ligand, i.e., L-cysteine in D_2_O (or NaOD/D_2_O), registered at three different pH (~7.0, ~11.0 and >13.0) (a–c), are compared with the spectrum of **R-Cys** (pH ~11.0) (d). At pH ~7.0, **Cys** is of zwitterionic nature; thus, its NMR spectra should be pH dependent. Indeed, with increasing pH, an increasing shift of all NMR signals is observed as a result of the deprotonation of the cationic part (-NH_3_^+^) of the zwitterion. This shift is caused by an increase in the electron density at the nitrogen atom, leading to more efficient shielding of all protons and their resonance at a higher field. Moreover, with increasing pH, the differentiation between diastereotopic protons of the methylene group in -CH_2_SH becomes more pronounced.

In the spectrum of **R-Cys**, no signals are observed in the spectral range 1.0–1.5 ppm, which implies absence of the initial ligands and confirms the efficacity of the ligand exchange process. However, if the spectrum of **R-Cys** is compared with the corresponding spectrum of free ***Cys*** registered and the same pH ~11, some differences in the chemical shifts of all protons can be noticed (compare Figure 5b,d). Moreover, the observed shifts are opposite to those induced by increasing pH. Thus, the binding of **Cys** to the nanocrystal surface results in electron density lowering in this molecule, decreasing shielding and resonance at the lower field.

As expected, the replacement of primary ligands in **R** (or **G**) by ***Cys*** to yield **R-Cys** (or **G-Cys**) resulted in a significant decrease in the content of carbon from 57.6% in **R** to 26.7% in **R-Cys** and from 55.3% in **G** to 24.6% in **G-Cys**. Ligand exchange also induced changes in the composition of the inorganic core, leading to a lowering of indium and zinc contents in **R-Cys** as compared to **R**. As in the case of **R-DHLA**, composition changes in the core of **R-Cys** were accompanied by a lowering of the nanocrystal size but had a rather limited effect on the emission band position. It was shifted from 720 nm for **R** to 707 nm for **R-Cys** (see Table 1 and Figure 2). This unexpected bathochromic shift accompanying decreasing nanocrystal size probably has its origin in two effects of opposite nature. Lowering the nanocrystal size induces a hypsochromic shift of the emission band due to the quantum confinement effect. To the contrary, diminution of the content of zinc results in a bathochromic shift of this band. The resultant weak bathochromic shift measured in **R-Cys** as compared to **R** must be considered as the product of these two opposite effects. The PLQY value drops from 67% in **R** to 28% in **R-Cys**.

**G-Cys** is characterized by the most pronounced change in the chemical composition of the inorganic core as compared to **G** which involves diminution in the content of silver and zinc. Since N is present only in **Cys** in the atomic ratio N/S = 1, the content of the “organic” sulfur can be estimated on the basis of nitrogen determination. This makes possible the estimation of the content of “inorganic”, core-originating sulfur by subtracting the content of “organic” sulfur from the total content of sulfur. The calculated molar ratio of silver to core-originating sulfur is 6.8 (**R-Cys**) and 38.7 (**G-Cys**). In both cases, these ratios are higher than those calculated assuming stoichiometric ratios of cations and anions. This means that the nanocrystals surface is enriched in anions in these cases. The discussed profound changes in the core composition of **G-Cys** as compared to **G** had only a weak effect on the position of its emission band which was slightly hypsochromically shifted from 543 nm in **G** to 524 nm in **G-Cys**. However, its PLQY value was significantly reduced from 48% to 16%.

## 4. Conclusions

To summarize, we have demonstrated that the exchange of hydrophobic primary ligands for hydrophilic ones in quaternary alloyed AgInS_2_-ZnS nanocrystals of varying composition can be relatively easily performed in a one-step procedure. This facile and efficient procedure cannot be applied to nanocrystals which contain strongly bound hydrophobic thiols as primary capping ligands such as DDT, for example. We have demonstrated that labile primary ligands, e.g., stearate anions co-ordinated by zinc cations and 1-aminooctadecane molecules co-ordinated by In(III) cations, are being readily exchanged for hydrophilic ligands containing thiol co-ordinating groups. Detailed characterization of nanocrystals in pre- and post-ligand exchange states has revealed that the exchange has a significant effect on the composition of nanocrystals, including the composition of their cores, and a much weaker effect on the nanocrystal size. These ligand exchange-induced composition and size changes have not affected the nanocrystals’ photoluminescence color to a large extent but have led to a significant diminution of the PLQY values. Among the three hydrophilic ligands tested—11-mercaptoundecanoic acid (MUA), L-cysteine (Cys) and dihydrolipoic acid (DHLA)—DHLA turned out to be the most promising since it assured the complete removal of the primary ligand and relatively high PLQY values after the ligand exchange: 40% for nanocrystals emitting red light and 28% for those emitting green light.

## Data Availability

The data presented in this study are available on request from the corresponding author.

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
