# Peer review of "Organic-to-Aqueous Phase Transfer of Alloyed AgInS2-ZnS Nanocrystals Using Simple Hydrophilic Ligands: Comparison of 11-Mercaptoundecanoic Acid, Dihydrolipoic Acid and Cysteine"

_nanomaterials, 2021, doi:10.3390/nano11040843_

Round 1
Reviewer 1 Report
This paper entitled Organic-to-Aqueous Phase Transfer of Alloyed AgInS2-ZnS 2 Nanocrystals Using Simple Hydrophilic Ligands. Comparison of 11-mercaptoundecanoic Acid, Dihydrolipoic Acid and Cysteine by Patrycja Kowalik, Piotr Bujak, Mateusz Penkala and Adam Pron describes the synthesis and characterization of AgInS2-ZnS nanocrystals differing in composition by ligands exchange that induces hydrophilization and differences in composition and by consequence emitting light of different colours: red and green.
The knowledge for this chemistry have been reported for the author in previous similar works; From Red to Green Luminescence via Surface Functionalization. Effect of 2-(5-Mercaptothien-2-yl)-8-(thien-2-yl)-5-hexylthieno[3,4-c]pyrrole-4,6-dione Ligands on the Photoluminescence of Alloyed Ag-In-Zn-S Nanocrystals. Inorganic Chemistry, 2020, doi: 10.1021/acs.inorgchem.0c02468,
The work is carried out in a good way and all compounds or materials are well characterized. The synthesis and characterization are clearly described. The article explains the synthesis procedure and the exhaustive characterization of the nanocrystal obtained by different techniques including NMR spectroscopy that indicates the ligand exchange is produced. Conclusions are good supported by the results.
I recommended this manuscript for publication after various considerations.
- Line 215: “As judged from elemental analysis, ligands exchange induces a decrease in the content of indium”. Moreover line 216; “For G-MUA a drop in the content of zinc is additionally observed.”. This fact could be produced by the lixiviation on indium or zinc cations Have you consider this fact for future applications indeed if they are biomedical applications? Thiols groups are presented in human proteins.
- Considerations of the recyclability could be interesting. There are not studies made about this fact: could be possible the ligand exchange to make hydrophobic crystals again?
- Typographic mistakes; line 42 are repeated words, line 95 mmola instead of mmol.
I recommended this manuscript for publication if these considerations are reviewed.
Thank you so much
Author Response
Assistant Editor Nanomaterials
Professor Johnny Wang
Dear Editor,
Please find enclosed a copy of the revised version of our paper entitled “Organic-to-Aqueous Phase Transfer of Alloyed AgInS2-ZnS Nanocrystals Using Simple Hydrophilic Ligands. Comparison of 11-mercaptoundecanoic Acid, Dihydrolipoic Acid and Cysteine” by Patrycja Kowalik, Piotr Bujak, Mateusz Penkala and Adam Pron (Manuscript ID: nanomaterials-1146326). Below you will find detailed answers to the referees’ comments.
Reviewer: 1
General comment:
“The work is carried out in a good way and all compounds or materials are well characterized. The synthesis and characterization are clearly described. The article explains the synthesis procedure and the exhaustive characterization of the nanocrystal obtained by different techniques including NMR spectroscopy that indicates the ligand exchange is produced. Conclusions are good supported by the results.” … “I recommended this manuscript for publication if these considerations are reviewed.”
Specific comments of Reviewer 1 are follows.
Comment 1.
“Line 215: “As judged from elemental analysis, ligands exchange induces a decrease in the content of indium”. Moreover line 216; “For G-MUA a drop in the content of zinc is additionally observed.”. This fact could be produced by the lixiviation on indium or zinc cations Have you consider this fact for future applications indeed if they are biomedical applications? Thiols groups are presented in human proteins.”
Our answer
This change in the nanocrystals core composition is caused by release of surfacial metal cations in highly basic conditions of the ligand exchange procedure. The exchange process is followed by nanocrystals precipitation and redispersion in water. At the final stage of purification highly diluted dispersions of nanocrystals in different buffers solutions of are obtained, which show significantly lower pH. In these conditions nanocrystals capped with hydrophilic ligands as well as those hybridized with biomolecules are stable.
Comment 2.
Considerations of the recyclability could be interesting. There are not studies made about this fact: could be possible the ligand exchange to make hydrophobic crystals again?
Our answer
So far we did not carry out recyclability investigations. However, it seems that this procedure could be difficult, at least for nanocrystals described in this research. Nanocrystals before the exchange of primary ligands contained no thiol-type anchor groups. This strongly facilitated the exchange process. In a reverse process of hydrophobization no primary ligands could be used but rather specially selected active hydrophobic thiols. As demonstrated by several reports, including ours on CuInS2-ZnS nanocrystals (G. Gabka et al. Phys. Chem. Chem. Phys., 2014, 16, 23082, G. Gabka et al. J. Phys. Chem. C, 2015, 119, 9656-9664), in the case of ligands with thiol anchor groups direct exchange is not possible (or at least very difficult). A two-step procedure is required in these cases involving pyridine labile ligands.
Comment 3.
Typographic mistakes; line 42 are repeated words, line 95 mmola instead of mmol.
These mistakes were corrected.
Reviewer: 2
General comment:
“The paper presents results on nanocrystals of AgInS2-ZnS (with different compositions) and modified by means of three different ligands exchange in a one-step procedure. Several analytical techniques were used for characterization, mainly NMR spectroscopy. Results show that the exchange depends mainly on the composition of nanocrystals, being independent of the size of the nanocrystals. The authors also report a significant diminution of the PLQY values.
manuscript is well written and shows important results in the field.”
Specific comments of Reviewer 2 are follows.
Comment 1
Some minor comments should be considered:
- i) Try to avoid the use of abbreviations in the abstract,
- ii) Introduction, define PLQY (photoluminescence quantum yield), which is defined in the abstract, but not in the text,
All these corrections were introduced.
iii) In this paragraph, which are the advantages/disadvantages of the two methods? Which are the biomedical applications? The authors seem to highlight these applications.
Following the referee’s request we added a short paragraph completing the description of the two strategies of preparation of hydrophilic nanocrystals. In the paragraph below the new text is indicated in blue.
“For many applications, including biomedical ones, nanocrystals of hydrophilic nature are needed, capable of forming stable aqueous colloidal solutions. Two strategies can be envisioned in the preparation of such nanocrystals [4,10]. In the first one, the nanocrystals are formed in aqueous solutions and their colloidal stability in this medium is assured by capping ligands of hydrophilic nature. The second one consists of preparing nanocrystals in non-aqueous media where they are stabilized with hydrophobic ligands. In the second step the exchange of hydrophobic ligands for hydrophilic ones is performed leading to the nanocrystals transfer to water, where they form stable dispersions. However, taking into account possible versatility of the surface modification, the second method seems more appropriate. This especially applies to hybrid systems frequently containing complicated bio-active molecules. The use of hydrophilic primary ligands severely limits the number of possible surface modification pathways as well as ligands exchange procedures [4]. Moreover, it sometimes makes the introduction of biologically active molecules difficult.”
- iv) Also in the introduction section, it seems that only NMR technique was the only one used to characterize the samples, but in the experimental section, other techniques are described and used.
Our new text is as follows:
“In this paper we comparatively describe ligands exchange-induced hydrophilization of two types of alloyed AgInS2-ZnS nanocrystals differing in composition and by consequence emitting light of different colors: red (R) and green (G). Three popular hydrophilic ligands were tested: 11-mercaptoundecanoic acid (MUA), dihydrolipoic acid (DHLA) and L-cysteine (Cys). Detailed characterization of nanocrystals and their surface was performed prior to the ligands exchange and after the exchange, which involved the determination of their size and shape, composition, crystal structure and luminescent properties. NMR was used for the identification of primary ligands and the ligands introduced upon the exchange”
Comment 2
Results, figure 1 in the supplementary information, which is the role of Oxygen?
Our answer
The identification of the nature of oxygen is difficult because it is present not only in all hydrophilic ligands but also in primary hydrophobic ones. Thus, unequivocal conclusions cannot be drawn from its content. Moreover, out earlier XPS studies focused on hydrophilic MUA ligands stabilizing CuInS2-ZnS nanocrystals (G. Gabka et. al. Phys. Chem. Chem. Phys., 2014, 16, 23082,), demonstrated the presence of water in the ligands sphere, which significantly modified the content of oxygen.
Comment 3
Figure S1. The X-ray emission for In-L and Ag-L are quite close, please the position for In should be clearly identified in the figure. In this image (red spectra), the intensity of Ag-L is clearly higher than the In-L, how it is possible to obtain this composition: Ag1.0 In3.1 Zn 1.0 S 4.00 (S6.15)? In one of the references, 24, in the supplementary information, a similar figure of EDS can be found. The composition for the nanocrystal is also similar: Ag1.0 In3.1 Zn 1.0 S 4.1 (S5.1), but in the EDS spectrum, the Ag-L/In-L ratio is completely different from that shown in this article. Check if the EDS spectrum used in this manuscript is the correct one, if not, the composition of the nanocrystal should be re-calculated. Consider also, in table 1, the precision of the EDS technique.
Our answer
The referee is correct. The spectrum in Figure S1 does not correspond to the formula. We mistakenly presented a spectrum recorded for a different sample. In the revised version Figure S1 in Supplementary Materials was corrected and the corresponding corrections in the main text were also introduced (table 1, page 4, 9 and 10).
To summarize, we addressed all comments of both referees. In addition, their comments enabled us to significantly improve our paper. We think that in its present form it can be accepted for publication in Nanomaterials.
Looking forward to hearing from you
Sincerely yours
Piotr Bujak

Reviewer 2 Report
The paper presents results on nanocrystals of AgInS2-ZnS (with different compositions) and modified by means of three different ligands exchange in a one-step procedure. Several analytical techniques were used for characterization, mainly NMR spectroscopy. Results show that the exchange depends mainly on the composition of nanocrystals, being independent of the size of the nanocrystals. The authors also report a significant diminution of the PLQY values.
The manuscript is well written and shows important results in the field.
Some minor comments should be considered.
- Try to avoid the use of abbreviations in the abstract.
- Introduction, define PLQY (photoluminescence quantum yield), which is defined in the abstract, but not in the text.
- Introduction, line 42, remove one “is performed”.
- In this paragraph, which are the advantages/disadvantages of the two methods? Which are the biomedical applications? The authors seem to highlight these applications.
- Also in the introduction section, it seems that only NMR technique was the only one used to characterize the samples, but in the experimental section, other techniques are described and used.
Results, figure 1 in the supplementary information, which is the role of Oxygen?
Regarding the data from EDS and corresponding spectra.
Figure S1. The X-ray emission for In-L and Ag-L are quite close, please the position for In should be clearly identified in the figure.
In this image (red spectra), the intensity of Ag-L is clearly higher than the In-L, how it is possible to obtain this composition: Ag1.0 In3.1 Zn 1.0 S 4.00 (S6.15)?
In one of the references, 24, in the supplementary information, a similar figure of EDS can be found. The composition for the nanocrystal is also similar: Ag1.0 In3.1 Zn 1.0 S 4.1 (S5.1), but in the EDS spectrum, the Ag-L/In-L ratio is completely different from that shown in this article.
Check if the EDS spectrum used in this manuscript is the correct one, if not, the composition of the nanocrystal should be re-calculated.
Consider also, in table 1, the precision of the EDS technique.
Author Response

(The authors gave the same response as above.)
